# Coffee Bean Characterization Using Terahertz Sensing

**DOI:** 10.3390/s25072096

**Published:** 2025-03-27

**Authors:** Dook van Mechelen, Daan Meulendijks, Milan Koumans

**Affiliations:** Department of Electrical Engineering, Eindhoven University of Technology, 5600 MB Eindhoven, The Netherlands

**Keywords:** optical properties of coffee beans, moisture level of coffee beans, coffee bean sorting, THz spectroscopy

## Abstract

Coffee bean sorting is currently based primarily on visual appearance and near-infrared techniques that probe the bean’s skin. However, sorting based on compositional differences has significant potential to optimize the roasting process. We present a novel coffee bean sorting method using terahertz (THz) spectroscopy, which effectively penetrates both green and roasted beans. Our findings show that the optical properties of coffee beans at THz frequencies are primarily governed by internal moisture levels. To demonstrate industrial feasibility, we implement a robot-guided THz sensing system capable of scanning beds of beans for automated sorting. More broadly, our results confirm the potential of THz technology for moisture content analysis across various applications.

## 1. Introduction

Coffee is a globally significant commodity, impacting both society and the economy. Optimizing industrial coffee processing is crucial, as even minor efficiency improvements can yield substantial savings due to the scale of the related industry. Large roasters typically source green coffee beans from a vast network of smallholder farmers, ensuring risk mitigation and consistent flavor profiles. While farmers may perform initial sorting, such as separating debris from beans, the roasting industry often applies additional selection criteria before processing. Current sorting technologies primarily rely on visible light inspection, which, when combined with machine learning, effectively classifies beans based on visual characteristics [1,2]. However, sorting based on bean composition remains a challenge. One key parameter is moisture content, which affects pricing, shelf life (due to mold growth), and roasting behavior [3]. The most accurate method for determining moisture is near-infrared (NIR) spectroscopy [4]. However, its limited penetration depth of only tens of microns means it primarily probes the bean’s outer skin [5]. The inferred moisture level is therefore assumed to represent the entire bean—an assumption that may not always hold for an inhomogeneous, organic product. In contrast, capacitive and microwave measurements probe the entire bean but with lower absolute accuracy. The potential of THz spectroscopy, having longer wavelength than NIR, was recently explored for ground coffee [6,7]. However, these studies do not provide insights into its applicability for coffee bean sorting, as powdered samples are substantially different from beds of whole beans.

Here, we explore THz transmission and reflection spectroscopy to analyze the composition of green and roasted coffee beans. By accurately determining the optical THz properties of both bean slabs and whole beans, we find that the dispersion of the complex refractive index exhibits characteristic signatures of moisture absorption, varying distinctly with roast level. Additionally, the penetration depth at 0.5 THz is approximately 150 μm, substantially greater than that of NIR radiation. We also test our method on immature quaker beans as an example of undesirable beans requiring removal during sorting. To evaluate the practical applicability of these findings in an industrial context, we have developed a robot-assisted THz sensing system capable of scanning clusters of coffee beans, demonstrating its potential for automated sorting.

## 2. Materials and Methods

THz time-domain spectroscopy (Teraflash, Toptica) was employed in both transmission and reflection geometries on whole coffee beans and thin slabs derived from them. Both setups utilize off-axis parabolic mirrors in a 4f configuration, and have a beam waist of approximately 3 mm at 1 THz. In transmission (reflection) geometry, the collimating mirrors have a focal length of 2″ (3″), while the focusing mirrors have a focal length of 6″ (7.5″). The shorter focal length of the collimating mirrors in the transmission setup facilitates the propagation of very low THz frequencies, reliably down to 0.03 THz, whereas in reflection, this limit is closer to 0.1 THz. The THz beam primarily contained frequencies between 0.1 and 2.0 THz, though signals with absolute error bars of ±0.5% were recorded up to 3 THz. Measurements were conducted under ambient conditions at 23 °C with relative humidity ranging from 20 to 35%.

Green Arabica coffee beans (Brazil) were acquired from Bocca Coffee Roasters (Netherlands). Roasted beans were manually prepared by heating green beans to 180 °C for three minutes until the first crack. An image of the beans is shown in Figure 1a. Given the bean thickness of 3.5–4.5 mm and transmission levels well below 1%, we also analyzed THz properties using slabs of coffee beans. To obtain these slabs, whole beans were ground from the flat (more porous) side, containing the characteristic groove, using fine sandpaper. The polished surface was hard and shiny, indicating that the polishing procedure did not alter the bean’s structure. The resulting slabs, being the denser part of the bean, had a thickness of 1.5 ± 0.05 mm.

Quaker beans were identified by splitting a large number of green beans in half. The other half was then roasted and visually inspected. Beans that exhibited a pale appearance after roasting were classified as quakers, accounting for approximately 10% of the total roasted beans. Given the significant visual difference from roasted mature beans, we consider this a reliable method for distinguishing quaker beans. The unroasted halves were retained as green quaker beans and were thinned down using the same method to obtain green quaker slabs.

Due to the irregular shape of coffee beans, careful consideration is required when determining their material properties. In transmission measurements, positioning a bean at the beam waist can introduce diffraction effects, as its size is comparable to the waist diameter, and the probed wavelength is of the same order as the structures on the bean. Additionally, coffee beans have a smooth, convex side and a rough but flat side with a longitudinal groove. The flat side can be directed towards the emitter, as orienting the convex side toward the emitter can create the effect of a convex mirror. To mitigate these issues, each bean was placed with its flat side against a thin adhesive metallic aperture with a 5 mm clear diameter (Figure 1b), which is smaller than the beans. This setup ensured that all detected radiation first passed through the bean, preventing any from bypassing it. A reference measurement was taken using an empty aperture. For reflection measurements, beans were affixed by their convex side onto a metal rod mounted on a kinematic stage using Crystalbond^TM^ adhesive. The mount allowed precise alignment to maximize the reflected signal. Reference measurements were obtained by coating the exposed (flat) side of the bean with RS PRO Silver Conductive Lacquer, followed by drying with a gentle nitrogen flow.

## 3. Results

### 3.1. Optical Properties of Coffee Beans

#### 3.1.1. Time-Domain Transmission

Figure 2 displays the directly recorded THz pulses in transmission geometry. Panel (a) shows the transmitted electric field $E_{T}\left(t\right)$ through a 5 mm clear aperture. In addition to the main transient, a characteristic beating pattern appears at longer delays due to the absorption of ambient water vapor. The transmitted signals for a 1.5 mm green bean slab and a 1.5 mm roasted bean slab are shown in panels (b) and (c), respectively. The absence of significant peaks before or after the main transmission through the slabs confirms that the flexible aperture effectively prevents unwanted light from bypassing the sample. Panels (e) and (g) present $E_{T}\left(t\right)$ for whole green and roasted beans. Unlike the slab measurements, these signals, at the detection limit of the spectrometer, do not exhibit a single transient but rather a prolonged sequence of fluctuating field strengths. Due to the millimeter-scale wavelengths of THz radiation, a minor portion of the signal diffracts around the sample holder and reaches the detector. However, this stray radiation is of the same order of magnitude as the signal transmitted through the bean itself. To quantify and correct for this stray radiation, we measured the detector signal with the 5 mm aperture fully closed, denoted as $E_{S}\left(t\right)$ (Figure 2d). This stray signal remained highly stable over time and exhibited strong correlations with features in $E_{T}\left(t\right)$ recorded from whole beans. By subtracting $E_{S}\left(t\right)$ from $E_{T}\left(t\right)$ we obtained the corrected transmission signals shown in Figure 2f,h. These now primarily feature two distinct transients, similar to the slab measurements. This correction is crucial because, without it, spurious low-frequency transmission bands appear in the power spectrum $T\left(\omega \right)={\left|E\left(\omega \right)\right|}^{2}$ after Fourier transformation, regardless of the actual sample properties.

#### 3.1.2. Transmission of Coffee Bean Slabs

$E_{T}\left(t\right)$, corrected for the stray radiation, was Fourier transformed to obtain $E_{T}\left(\omega \right)$. To verify the stability of the transmission setup, we took ratios of clear-aperture transmissions $E_{0,T}\left(\omega \right)$ taken at different time intervals (Figure 3a,f). These so-called noise levels demonstrated high reproducibility, with variations within ±0.2% above 0.1 THz. The complex sample transmission was determined as $t\left(\omega \right)=E_{T}\left(\omega \right)/E_{0,T}\left(\omega \right)$, and the corresponding transmission intensity was $T\left(\omega \right)={\left|t\left(\omega \right)\right|}^{2}$. These quantities are presented in Figure 3 for both coffee bean slabs (panels a–e) and whole beans (panels f–j). Measurements were performed on 25 to 30 individual samples per condition. Given the structural inhomogeneity of coffee beans and the high degree of polarization of the used THz radiation, spectra were recorded at varying mutual orientations to account for these factors. Individual spectra exhibited substantial variation due to sample inhomogeneity, as indicated by the error bars in Figure 3. At equal thicknesses, roasted bean slabs showed significantly higher transmission than green bean slabs. On a logarithmic scale, T(ω) of roasted slabs exhibited a dome-like transmission window between 0.05 and 2 THz. In contrast, green bean slabs showed a similar spectral shape but with transmission extending only up to approximately 1.5 THz. Interestingly, the transmission spectrum of green bean slabs (Figure 3a) exhibited a local minimum around 0.65 THz. While studies on THz properties of organic compounds may attribute such features to vibrational lattice modes, this explanation does not hold here. Instead, such dips arise from structural inhomogeneities, where THz waves interact with distinct regions of the sample exhibiting slightly different optical properties. This scenario is analogous to a birefringent crystal, where a light wave polarized at an intermediate angle between two optical axes experiences a mixed response. Visually, the central region of a coffee bean slab, including its side lobes, appears different from the surrounding material. Our THz measurements confirm that its optical properties also differ significantly, as discussed later (see Section 3.3). Even small differences in optical properties between regions can result in two effective beams transmitting through the sample which interfere with each other, leading to spectral dips. In agreement with the expectation that coffee beans exhibit low crystallinity, we did not observe sharp spectral signatures.

#### 3.1.3. Transmission Profile Correlated to Coffee Bean Constituents

To explain the overall dome-like shape of T(ω) determined on coffee bean slabs, we consider the primary constituents of coffee beans. Green beans contain approximately 10–12% water, whereas light roasting reduces this content to about 4–5%. Additionally, the 2–10% sucrose present in green beans is entirely converted upon roasting [8]. The transmission spectrum of thick liquid water films also exhibits a dome-shaped profile, peaking around 0.25 THz, with a steeper decline at lower frequencies [9]. Furthermore, the absorption of sucrose steadily increases above 1 THz, with a pronounced peak at 1.8 THz [10]. However, the absorption peak of sucrose in coffee beans may be significantly broadened due to the lack of crystallinity. Based on these observations, we attribute the transmission characteristics of the measured coffee bean slabs primarily to water absorption, with a possible secondary contribution from sucrose.

#### 3.1.4. Transmission Modeled Using Transfer Matrix Method

To characterize T(ω) and to model the measured dispersion, we fitted T(ω) using a transfer matrix method with a Lorentz parameterization, incorporating two Lorentz oscillators. We would like to emphasize that while these dispersion characteristics are commonly used to analyze crystalline solids, they may not necessarily be the best-suited choice for organic products. To ensure accurate fitting of the phase, we simultaneously fitted T(ω) along with both the real and imaginary parts of t(ω), i.e., Re t(ω) and Im t(ω) [11]. We opted not to fit $E_{T}\left(t\right)$ directly, as our primary objective was to model frequency-domain features such as the dome-shaped profile of T(ω), which are less straightforward to evaluate in the time domain. The fitting results for green and roasted bean slabs are presented in Figure 3a–e. One Lorentz oscillator was placed below the spectral range, near DC, to capture the sharp decrease in T(ω) below 0.3 THz. The second oscillator was positioned in the higher THz range, outside the measured window, to account for the gradual decline of T(ω) above 0.3 THz. Since these oscillator frequencies lie outside the measurement window, we do not focus on their precise parameter values but rather on the resulting dispersion characteristics. The complex index of refraction n(ω), determined from transmission measurements on the bean slabs, is shown in Figure 3k. Both Re n(ω) and Im n(ω) are slightly higher for green beans than for roasted beans, consistent with the observed transmission differences. Notably, the sharp increase in Im n(ω) at low frequencies corresponds to the strong decrease in T(ω) below 0.3 THz.

#### 3.1.5. Transmission of Whole Coffee Beans

We repeated the experiment with whole beans, again orienting their flat side toward the emitter to maximize the detected transmitted radiation. The overall transmission profile of whole beans aligns well with expectations based on T(ω) of coffee bean slabs, adjusted for their larger thickness. This confirms the reliability of our whole-bean measurements despite the increased experimental challenges. For green beans, consistent with $E_{T}\left(\omega \right)$, T(ω) remains very small, well below 0.5%. Notably, for roasted beans, T(ω) exhibits a local minimum around 0.8 THz, similar to the previously discussed dip in green slab measurements. We attribute this feature to the more porous material on the flat side of the bean, which was removed to obtain the slab, which has slightly different optical properties than the remaining part of the whole bean. As with the slabs, we fitted T(ω) and t(ω) using the same model-based fitting method. The resulting n(ω) for whole green and roasted beans is shown in Figure 3l. While whole beans contain additional material that may slightly differ from the slabs, evidenced by the extra 0.8 THz dip in roasted beans, we expect their optical properties to be comparable. Indeed, a comparison between Figure 3k,l reveals highly similar optical properties for roasted beans. For green beans, the spectral window where transmission remains reliable, i.e., for $T\left(\omega \right)>10^{-7}$, is significantly narrower than that of the bean slabs. As a result, the optical properties derived from whole green beans are somewhat less reliable than those from green bean slabs. Nonetheless, the match between n(ω) for whole beans and slabs within the 0.1–0.8 THz range is acceptable for an organic compound. The overall agreement between the optical properties of slabs and whole beans underscores the reliability of our measurements. This is particularly noteworthy given the inherent challenges of working with organic materials, which often exhibit significant inhomogeneity and irregular shapes.

#### 3.1.6. Reflectivity of Coffee Beans

The previous analysis examined the THz characteristics of coffee beans using transmission geometry. However, a reflection geometry would be more closely aligned with industrial sorting applications. This study was conducted exclusively on whole beans, as their shape directly influences the recorded reflectivity. Figure 3m presents the reflectivity R(ω) of green and roasted beans, revealing that green beans reflect slightly more than roasted beans. This aligns with expectations based on n(ω) determined from transmission measurements. To further analyze the structure in R(ω), distinguishing between noise and actual material responses, we computed $\hat{R}\left(\omega \right)$ of a whole bean using the previously determined n(ω) of coffee bean slabs (Figure 3n). Consistent with the transmission profiles, $\hat{R}\left(\omega \right)$ increases sharply below 0.2 THz due to the large real part of the refractive index Re n(ω), which originates from the GHz dielectric relaxation of water [12]. Above 1 THz, $\hat{R}\left(\omega \right)$ decreases gradually without prominent spectral features. Between these frequency ranges, we observe a narrow spectral region where Fabry–Pérot oscillations appear. Interestingly, a similar behavior is evident in the experimental reflectivity curves R(ω). Despite the inhomogeneous composition and irregular shape of whole beans, the interference pattern remains clearly visible. Given that coffee beans are relatively small compared to the probed wavelengths and the beam waist, accurately characterizing them in reflection is challenging. Nonetheless, our transmission and reflection measurements provide a coherent and consistent picture of the THz properties of coffee beans.

#### 3.1.7. Relevance of Optical Properties of Coffee Beans for Industrial Applications

We would like to comment on the practical utility of the optical properties determined in this study, besides providing insights into the structure of a coffee bean. When considering an industrial coffee bean sorting application, the prevailing approach today relies on conventional AI methods for classification, often without a deep understanding of the underlying spectral differences between beans that need to be differentiated [6,13]. However, large coffee producers, who would benefit from sorting beans before roasting, source their beans from farms across the world. Successfully deploying AI methodologies without training on possible variations among beans has proven to be challenging [14]. To overcome this, a more robust approach is to train AI models on well-understood spectral features that reliably distinguish between different bean types. This enhances the model’s generalizability and ensures it remains effective even when new or untrained samples appear. Furthermore, hybrid AI schemes where data-driven learning is combined with physics-based modeling offer an inherently more adaptable solution [15]. In these approaches, well-characterized optical properties play a crucial role in feature selection and model performance. For these reasons, accurately determining the optical properties of coffee beans is not just of academic interest but is also of significant practical importance for improving real-world sorting applications.

### 3.2. Optical Properties of Quaker Beans

Quaker beans are among the undesirable beans that should be removed before roasting. Green quaker beans are immature beans that lack sucrose, preventing them from developing the same sensory attributes as mature green beans when roasted. Figure 4 presents the transmission spectra of 0.5 mm thick green quaker bean slabs compared to 0.5 mm thick green mature bean slabs. The error bars indicate the variability observed across multiple measurements. Please note that the noise level is higher than in previously displayed data (Figure 3). This is because the bean slabs are much smaller, as they are split in the green state (see Section 2). Therefore, an aperture of 2 mm was used. While the overall transmission profile of green quaker beans follows the same trend as that of green mature beans, their transmission is up to 20% higher. Although detailed compositional data on quaker beans is limited, it is well established that their gravimetric density is lower. The increased transmission in green quaker bean slabs is likely due to this lower density, which results in reduced overall moisture content. The featureless absorption dispersion of sucrose in the spectral range of T(ω) combined with small differences in transmission, hinders the reliable determination of sucrose level differences between the two green bean types.

### 3.3. Terahertz Tomography of Coffee Beans

To better understand the compositional inhomogeneity of a coffee bean, we conducted a raster scan of 32×32 pixels in transmission geometry on a roasted coffee bean. As in previous experiments, the bean was sealed within a metallic aperture to prevent radiation from bypassing the bean and reaching the detector (Figure 5). THz transmission images were generated by plotting $\int |E_{T}{\left(\omega \right)|}^{2}d\omega$ for frequency bins between 0.2 and 1.2 THz (Figure 5). These measurements provide insights into the spatial variation of the THz properties within the coffee bean. Please note that incorporating information at higher frequencies decreases the spot size, allowing for greater detail in the corresponding image. Since no transmission occurs outside the oval aperture, the borders of each image appear as dark blue, forming an oval shape around the yellow areas. A common feature observed across all frequency bins is the presence of two distinct side lobes. Additionally, a vertical groove with lower transmission is consistently visible. Despite the reduced material in this area, the lower transmission could be attributed to the presence of the silver skin. Notably, in the highest frequency bin, the embryo becomes distinctly visible as a small circular region at the top of the image (indicated by the arrow). These results confirm the previously assumed inhomogeneity of coffee beans. A THz beam passing through a whole coffee bean simultaneously interacts with regions of differing optical properties, leading to the observed dips in T(ω) (see Figure 3a,f).

### 3.4. Toward the Industrial Sorting of Coffee Beans Using THz Spectroscopy

So far, we have focused on obtaining accurate properties of coffee beans by considering their alignment, straylight effects, and proper references. However, for industrial applications, coffee beans will pass by in large numbers, and for inspection, individual signatures need to be recognized when probing an entire batch. To explore THz technology for the industrial sorting of coffee beans, we have built an appropriate demonstrator setup. We use a less costly frequency-domain THz spectrometer (T-sweeper, Toptica), with photoconductive antennas integrated into a lens-based reflection head (Figure 6a,b). The THz beam is slightly focused with a diameter of about 10 mm to probe several beans at the same time. The measurement head contains laser-distance sensors and is mounted on a robot (Kinova Jaco 2) (Figure 6a). The sensors are used to determine the position and orientation of the head relative to the inspected surface. The robot is controlled using the Robot Operating System (ROS) and the MoveIt Motion Planning Framework. We prepared a pattern of coffee beans consisting of green and roasted beans, as well as the university logo TU/e made from 3D-printed plastic (Figure 6c). The robot aligns the optical head perpendicular to the surface and maps the sample pattern along a scanning trajectory. From the mapped integrated THz intensity in a narrow frequency range between 0.2 and 0.8 THz, we clearly recognize the TU/e logo (Figure 6d). The green beans show up as strongly reflecting THz radiation, more pronounced than the roasted beans. As the load of the reflection head, including connecting cables, is relatively large for the robot, the position accuracy decreases at the end of each path. This can, for instance, be seen in the right part of the image, where the ‘e’ in TU/e and the beans are not well distinguishable. We provide further insight into the reflection of coffee bean beds by inspecting R(ω) of green and roasted beans in the scanned pattern. The reflectivity is more than an order of magnitude lower than that measured in a well-aligned bean (Figure 3m). The frequency-domain spectrometer has a much smaller dynamic and spectral range compared to the time-domain spectrometer used previously. Additionally, taking an appropriate reference is difficult, as a flat mirror is certainly not the best option. However, painting the entire bed with metallic paint, as done for an individual bean, is not compatible with the application. Despite the large spread, a clear statistical difference can still be observed between green and roasted beans.

## 4. Discussion

We conducted a comprehensive study on determining the THz optical properties of coffee beans as a function of roast level. The dispersion of the complex refractive index, n(ω), is primarily influenced by moisture content. Unlike near-infrared radiation, which is commonly used to determine moisture content, THz radiation offers a significantly greater penetration depth, δ=150μm around 0.5 THz. Despite this advantage, it remains uncertain whether the moisture level obtained from such measurements accurately represents the entire bean. Therefore, we evaluated moisture content on two length scales by comparing the optical properties deduced from transmission measurements with those from reflectivity measurements. We first determined n(ω) based on both a bean slab and a whole bean. Subsequently, we recorded the reflectivity of whole beans and compared it with the predicted reflectivity based on n(ω). The strong agreement between these results, particularly the water-like dispersion, suggests that a penetration depth of 150 μm is sufficient for assessing overall moisture content via reflectivity measurements. We conclude that THz technology is a promising candidate for probing the composition of green and roasted coffee beans.

To assess the potential of THz technology for industrial coffee bean sorting, we demonstrated reflectivity measurements on a bed of beans using a robot-guided setup. Similar to individual beans, green beans reflected more THz radiation than roasted beans, though the contrast was limited. Improving the signal-to-noise ratio is essential for industrial viability. This could be achieved by increasing the power of the photoconductive emitter, potentially enhancing the dynamic range by two orders of magnitude [16], and optimizing signal collection by using large-aperture mirrors placed close to the sample. Unlike many studies where machine learning is applied without feature identification, our results reveal clear spectral features related to moisture that could serve as robust inputs for classical machine learning algorithms. Additionally, a THz camera could enable large-scale inspection, making AI-based image processing more effective than pointwise scanning. In summary, the current limitations in detected intensity due to random bean orientation must be addressed before THz technology for coffee sorting can become industrially viable. However, the strong reflectivity contrast between green and roasted beans presents an opportunity for THz-based inspection. If optimized, THz technology could significantly outperform existing sorting methods based on near-infrared, microwave, and capacitance measurements.

We aim to place these results in a broader context, as THz technology is still establishing its role in societal and industrial applications. Our work reinforces the potential of THz technology for moisture sensing and demonstrates its practical application in the food sector, where organic materials exhibit complex shapes and structures. Recent studies have demonstrated the effectiveness of THz spectroscopy for determining moisture content in wood [17,18], plant leaves [19,20], plant surfaces [14], and paper sheets [21]. While THz technology has not yet proven competitive in the biomedical field, where most materials are water-rich, it is a strong candidate for materials with low water content. Our findings support this trend by demonstrating that, in food products, where moisture content is particularly important, THz spectroscopy provides an accurate and effective means of measurement.

## Figures and Tables

**Figure 1 sensors-25-02096-f001:**
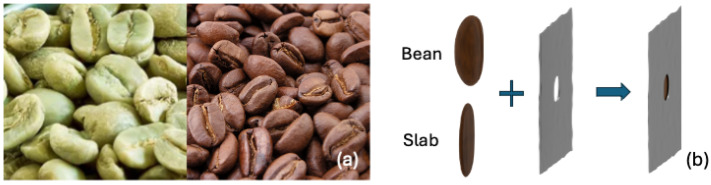
(**a**) Photograph of the used green and roasted coffee beans. (**b**) Procedure of how coffee bean samples have been prepared: by placing whole coffee beans or coffee bean slabs behind an adhesive flexible metallic sheet containing a 5 mm hole.

**Figure 2 sensors-25-02096-f002:**
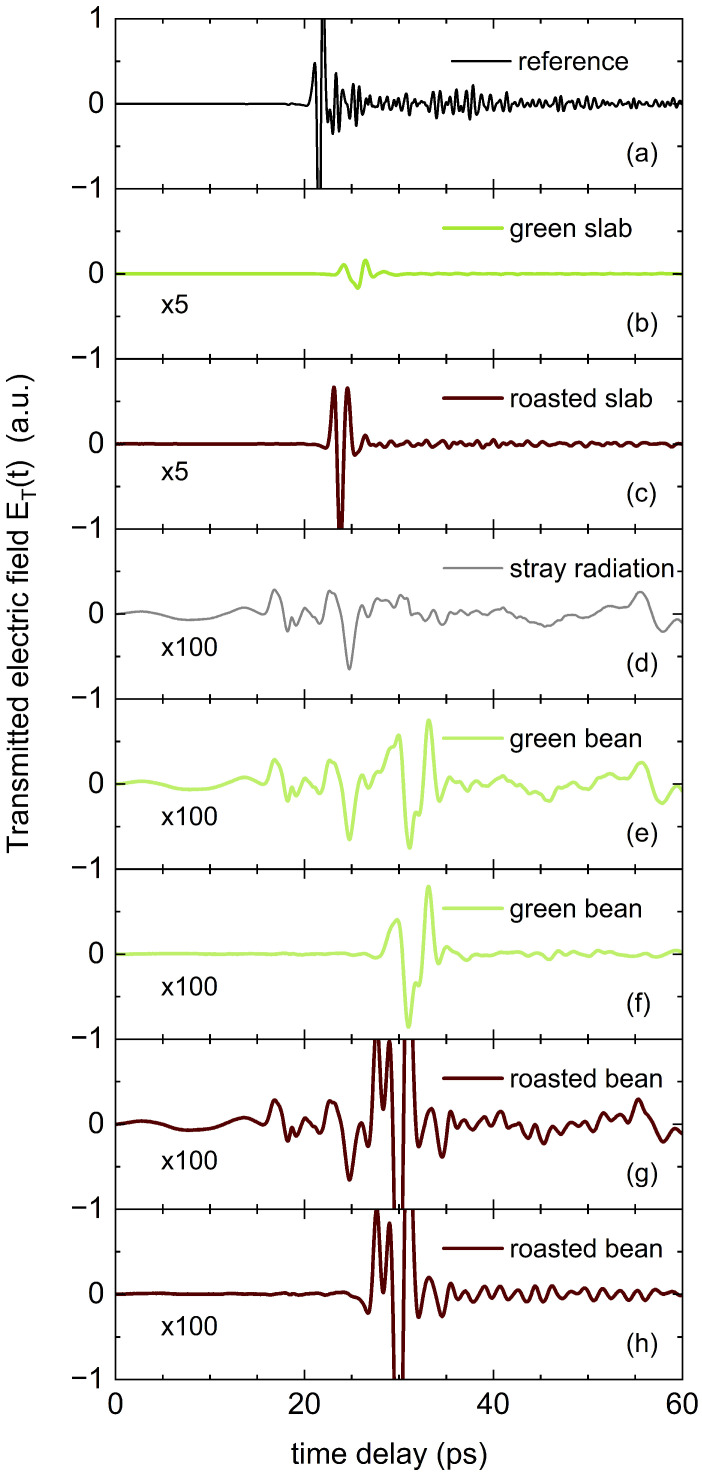
Experimental transmitted electric field of (**a**) reference measurement using an Al mirror, (**b**) 1.5 mm thick green coffee bean slab, (**c**) 1.5 mm thick roasted coffee bean slab, (**d**) background stray radiation radiation recorded by blocking the 5 mm aperture, (**e**) 3.5 mm thick whole green coffee bean, (**f**) corrected signal from (**e**) after subtracting (**d**), (**g**) 3.8 mm thick whole roasted coffee bean, and (**h**) corrected signal from (**g**) after subtracting (**d**).

**Figure 3 sensors-25-02096-f003:**
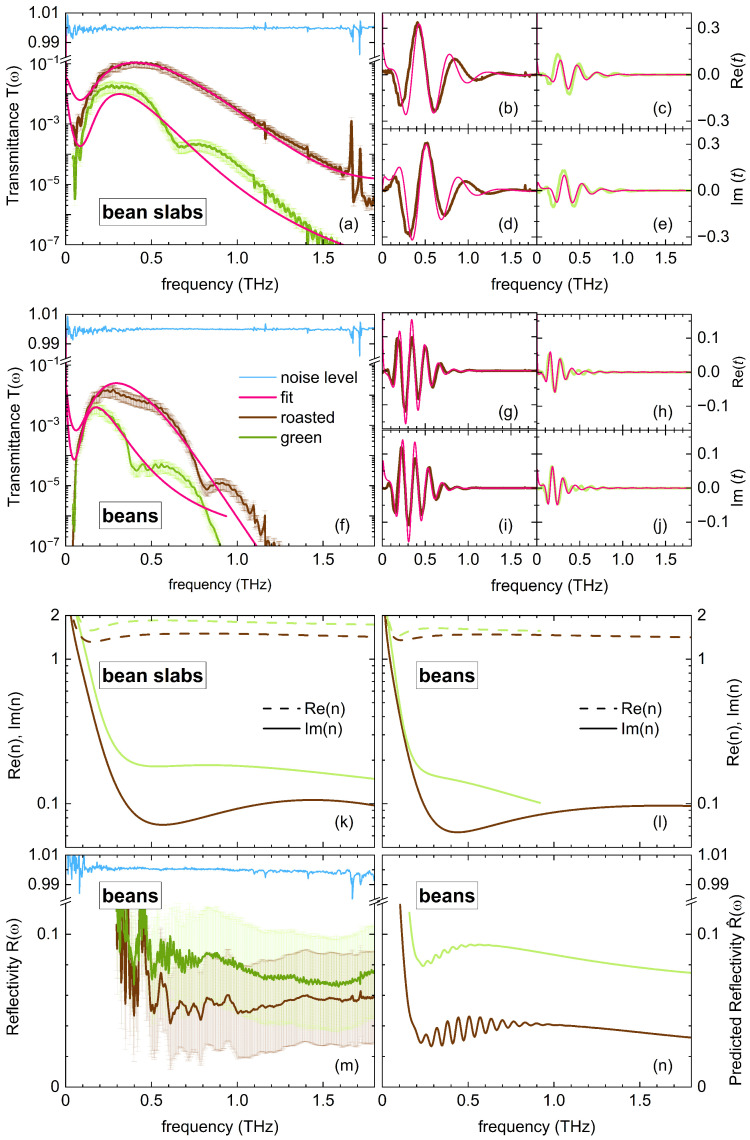
Optical properties of coffee beans at THz frequencies at ambient conditions. Error bars are estimated from measurements a wide range of coffee beans. (**a**) Transmittance T(ω) of green and roasted coffee bean slabs of 1.5 mm thickness. Also shown is the noise level of the experimental setup. (**b**,**c**) Corresponding Re t(ω) and (**d**,**e**) corresponding Im t(ω). (**f**) Transmittance T(ω) of green and roasted whole beans. (**g**,**h**) Corresponding Re t(ω) and (**i**,**j**) corresponding Im t(ω). Red lines are best fits based on a transfer matrix method (see text). (**k**) Real and imaginary part of the index of refraction n(ω) of green and roasted coffee beans slabs deduced from fits on data in panels (**a**–**e**). (**l**) Idem for whole green and roasted coffee beans deduced from fits on data in panels (**f**–**j**). (**m**) Reflectivity R(ω) of whole green and roasted beans. Also shown is the noise level of the experimental setup. (**n**) Reflectivity $\hat{R}\left(\omega \right)$ determined from n(ω) of coffee bean slabs displayed in panel (**k**).

**Figure 4 sensors-25-02096-f004:**
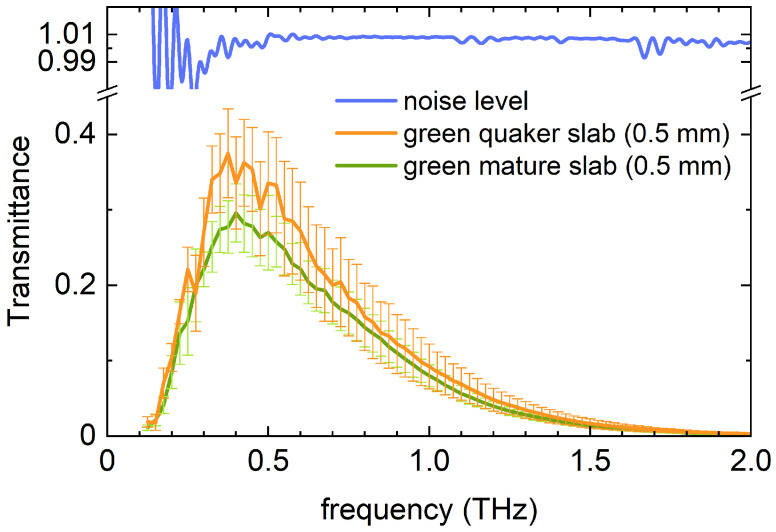
THz transmittance T(ω) of 0.5 mm thin bean slabs, for green quaker beans (orange) and green mature beans (green). Error bars are estimated from measurements on a wide range of beans. Also shown is the noise level of the experimental setup (blue line).

**Figure 5 sensors-25-02096-f005:**
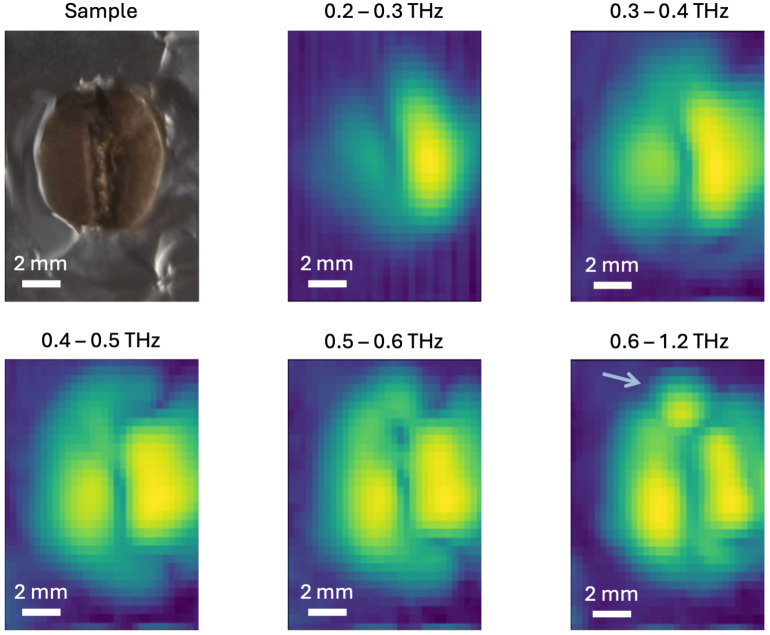
Photograph of a coffee bean surrounded by the metallic aperture (top left). THz transmission images (32×32 pixels), plotted as $\int |E_{T}{\left(\omega \right)|}^{2}d\omega$, of the coffee bean for the indicated frequency windows (dark blue is zero transmission and yellow is maximum transmission). The images reveal the sidelobes and the position and shape of the embryo (indicated by the blue arrow in the bottom right image).

**Figure 6 sensors-25-02096-f006:**
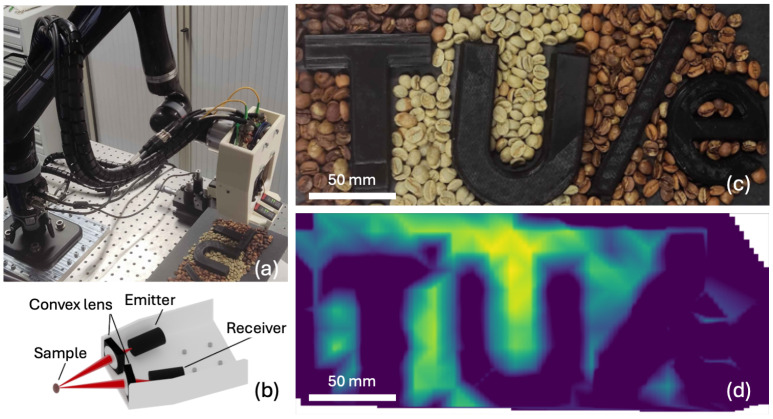
Demonstrator experiment showing (**a**) the experimental setup of a robot-mounted optical head scanning a pattern containing beds of green and roasted coffee beans, as well as a 3D-printed plastic university logo TU/e. (**b**) Schematic drawing of the optical head showing the two THz antennas (emitter and receiver) as well as the focusing optics. (**c**) Photograph of the pattern. (**d**) Integrated intensity of received THz radiation between 0.2 and 0.8 THz, showing the enhanced intensity for green beans.

## Data Availability

The datasets, as well as the developed code for analysis, can be made available upon reasonable request from the corresponding author.

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
