# Peer review of "Coffee Bean Characterization Using Terahertz Sensing"

_sensors, 2025, doi:10.3390/s25072096_

Round 1
Reviewer 1 Report
Comments and Suggestions for Authors
In Fig. 2, it looks as if the vertical scale goes from -1 to -1 in arbitrary units. Maybe scale the vertical axis to -1.2 to 1.2 so that the numbers from different sub-figures do not overlap.
In line 125, "rotational lattice modes" are mentioned. Should this have been "vibrational lattice modes"? If not, a reference would be needed here.
Fig. 3 sub-figure (n). The results presented are based on slabs, but are used to the figure bear the label "bean". It is also not clear from the text what the noise levels included in sub-figures a, m and f of fig. 3 represent. Is this the measured signal when the THz radiation is blocked?
Section 3.1.4: Without any motivation for the choice of fitting function used for the transmission data, the choice seems random. It is also unclear how the dip in the spectra, which is allocated to interference from internal reflections, influence the parameters extracted from the fit, which should somehow be addressed.
line 201, the reference to "the GHz dielectric relaxation of water" should be supported by a literature reference.
Fig. 5: It is unclear how this image was made. Is the bean placed on a 5mm metal aperture and raster scanned with a 3mm spot size THz beam? In that case the achieved spatial resolution is impressive, but do not show the expected effect of the beam being clipped at the edges, which might need to be commented on. If not, how was it done?
The robot assisted measurements rely only on the reflectivity contrast due to different water content, and the spectral differences within the scan-range (0.2-0.8 THz) are not really used for anything. This makes the link to the rest of the paper very weak. Please consider to either use the spectral information based on the studies in the first part of the paper or leave this part for a separate publication. The former could for example be done by subtracting images at two different frequencies (or frequency intervals to average out any interference fringes), where the relative reflectance of the green and roasted beams are different.
Author Response
- In Fig. 2, a ‘1’ is indicated on the upper axis, clearly marking the upper limit. We attempted the reviewer's suggestion; however, it resulted in a long list of ‘-1’ and ‘1’ stacked vertically, which diminishes the visual appeal. We therefore like to stick to the current representation, having all correct information.
- Rotational has been changed to vibrational.
- The reviewer correctly states that the choice of the dispersion relation (we believe that is what is meant with ‘fitting function’) was not motivated. Therefore, we have added in section 3.1.4: “We would like to emphasize that while these dispersion characteristics are commonly used to analyze crystalline solids, they may not necessarily be the best-suited choice for organic products.”
In relationship to fitting the dip, which is not due to internal reflections as the reviewer states (please see text: “This scenario is analogous to a birefringent crystal, where a light wave polarized at an intermediate angle between two optical axes experiences a mixed response.”) we mention in the text that, in addition to T, we also fit Re(t) and Im(t). Although the dip is clearly visible in T, it is not as apparent in the other two functions. We would like to emphasize that T is plotted on a logarithmic scale, which magnifies small variations. Our fit line remains accurate even on this log scale, demonstrating its accuracy.
- The reviewer correctly states that in Fig. 3n the label ‘bean’ has been used as the ‘predicted reflectivity’ indeed relates to a whole bean instead of a bean slab. To enhance clarity we added this explicitly in section 3.1.6:
To further analyze the structure in $R(\omega)$, distinguishing between noise and actual material responses, we computed $\hat{R}(\omega)$ of a whole bean using the previously determined $n(\omega)$ of coffee bean slabs (Fig.~\ref{fig3}n).
The origin of the error bars is explained under 3.1.2. However, to enhance clarity we there added the term ‘noise level’ as later used in the text:
$E_T(t)$, corrected for the stray radiation, was Fourier transformed to obtain $E_T(\omega)$. To verify the stability of the transmission setup, we took ratios of clear-aperture transmissions $E_{0,T}(\omega)$ taken at different time intervals Fig.~(\ref{fig3}a,f). These so-called noise levels demonstrated high reproducibility, with variations within $\pm0.2$\,\% above 0.1 THz.
- We have added an appropriate reference for the GHz relaxation of water: Lunkenheimer, P.; Emmert, S.; Gulich, R.; Köhler, M.; Wolf, M.; Schwab, M.; Loidl, A. Electromagnetic-radiation absorption by water. Physical Review E 2017, 96, 062607
- We thank the reviewer for the remark. As mentioned in the text, a bean is sealed inside a metallic aperture. As mentioned in the text: “THz transmission images where generated by plotting $\int|E_T(\omega)|^2d\omega$ for frequency bins between 0.2 and 1.2 THz (Fig.~\ref{fig5}).” (we corrected here t->\omega, as this was erroneously mentioned in the previous version of the manuscript). The spot size is frequency-dependent. Therefore, as the maximum frequency increases, the image reveals more detail. We have added this to the manuscript in section 3.3 as “Please note that incorporating information at higher frequencies decreases the spot size, allowing for greater detail in the corresponding image. Since no transmission occurs outside the oval aperture, the borders of each image appear as dark blue, forming an oval shape around the yellow areas.”
- The purpose of the robot-assisted measurements is to demonstrate the potential of THz technology for coffee bean sorting, as mentioned in Section 3.4. We believe these measurements provide a clear illustration that THz technology can distinguish coffee beans based on water content, consistent with all other sections of the manuscript. The statement of the reviewer ‘The robot assisted measurements rely only on the reflectivity contrast due to different water content, and the spectral differences within the scan-range (0.2-0.8 THz) are not really used for anything.’ is unfortunately not correct. The spectral differences (i.e., the dispersion of the optical properties, such as n(\omega)), in the THz range are primarily influenced by moisture in the coffee bean, which is the main message of the manuscript. Therefore, the water content is directly linked to the spectral properties measured in this demonstator experiment.
Reviewer 2 Report
Comments and Suggestions for Authors
The manuscript titled "Coffee Bean Characterization Using Terahertz Sensing" presents a novel coffee bean sorting method using terahertz (THz) spectroscopy. Furthermore, In my opinion, the paper is well-written and can be accepted with the following minor corrections:
- The citation in the text for a figure should appear before the figure itself. In fact, for example, Figure 1 is shown before its citation in the text (see lines 66-75). Please fix this point.
- Lines 325-327: This part of the manuscript should be added to the introduction as it discusses the potential of THz spectroscopy. When referring to the use of THz in plant leaves, please include this relevant reference:
DOI:1186/s13007-017-0197-z
3. A concluding paragraph would be appreciated.
Author Response
- We appreciate the reviewer's comment on formatting, which we have also noted. However, the LaTeX compilation automatically places the graph in that position, regardless of the commands in the text file. We suggest leaving this adjustment to the MDPI typesetters, who will ensure it aligns with the journal’s formatting standards.
- We thank the reviewer for their suggestion. However, as the paper does not pertain to THz in plant leaves nor serve as a review on that topic, we propose to keep the manuscript as it is.
- The journal policy states about a Conclusion section that ‘This section is not mandatory but can be added to the manuscript if the discussion is unusually long or complex.’ We believe that our discussion is neither unusually long nor complex. Therefore, we prefer to adhere to the journal's standard.
Reviewer 3 Report
Comments and Suggestions for Authors
- In sample preparation, coffee bean slabs are ground with sandpaper, which may damage the internal structure. How to ensure the optical properties of the slabs accurately represent those of the corresponding whole bean areas?
- Quaker beans are classified post-roasting by color, lacking direct chemical validation (e.g., sucrose content). How to ensure classification accuracy?
- In reflection measurements, the convex surface of coffee beans may introduce specular reflection or scattering effects, impacting the industrial applicability of the reflection signal.Please explain this.
- Is the scanning speed of the robot-guided THz system sufficient for industrial production line demands (e.g., hundreds of beans per minute)?
- In industrial settings, coffee beans may have dust or surface moisture. Do these interference factors affect THz measurements?
- The terahertz sensing technology is now progressing rapidly. Please refer to the relevant terahertz sensing literature and provide an evaluation of it,
All-Dielectric Metasurface-Based Terahertz Molecular Fingerprint Sensor for Trace Cinnamoylglycine Detection, Biosensors 2024, 14, 440;
Comments on the Quality of English LanguageNONE
Author Response
- We thank the reviewer for their comment. In fact, DvM had raised the same concern with the co-authors prior to the experimental work. However, we were surprised by how hard the interior of a coffee bean is and how remarkably smooth the surface becomes with just sandpaper. To share this experience with the reader, we have added to the Methods section 2: “The polished surface was hard and shiny, indicating that the polishing procedure did not alter the bean's structure.”
- The reviewer correctly notes, as mentioned in the manuscript, that quaker beans were sorted based on visual appearance after roasting. Chemical validation was not within the scope of this work. The visual difference between a quaker bean and a mature bean is significant. To emphasize this, we have added a clarification in the Methods section: “Given the significant visual difference from roasted mature beans, we consider this a reliable method for distinguishing quaker beans.” However, some beans may indeed have been misclassified. Likewise, chemical validation also includes error margins, and misclassification can occur in that process as well.
- The reviewer correctly notes that the geometrical properties of beans influence the detected signals. Therefore, we had included the demonstrator experiment mentioned in Section 3.4, where beans are randomly oriented, as shown in Fig. 6b. Despite variations in orientation, we are still able to distinguish beans based on their roast level.
- The current demonstration in Section 3.4 illustrates the concept. Further technology and product development is required to meet the requirement specifications of the relevant industries, which is beyond the scope of this work.
- In the demonstration in Section 3.4, the beans also have surface moisture and are not entirely dust-free. However, this does not pose an issue for the proposed method, as we can still distinguish beans by their roast level.
- We believe that we have included all relevant literature in the current manuscript. The manuscript is not related to ‘All-Dielectric Metasurface-Based Terahertz Molecular Fingerprint Sensor for Trace Cinnamoylglycine Detection’. Therefore we propose to leave the manuscript as it is.
Reviewer 4 Report
Comments and Suggestions for Authors
This work reports on a new coffee bean sorting method using terahertz spectroscopy, providing a completely new method for achieving coffee bean sorting. The authors mainly explore how to analyze the optical properties of coffee beans, especially the internal water content, by THz spectroscopy, and demonstrate its feasibility in industrial applications. This work improves the depth and accuracy of the beans, and unlike the past, requires grinding the whole beans. The paper is well written but there are still minor shortcomings to address and I recommend making minor modifications before accepting.
- In section 3.1.3, you discuss the change in the absorption peak of green coffee beans after baking, mentioning that the absorption summit of sucrose in coffee beans is wider due to the lack of crystallization. What are the specific influencing factors between the two?
- In section 3.2, you have discussed the transmission difference between green Quick beans and mature beans. If you use a 2mm aperture, can you suggest the specific reasons for choosing this aperture?
- You have proposed an AI model for screening coffee beans, but does it fit most of the actual measurements? If possible, it is hoped that enough test results can be supplemented to verify whether the model is useful.
- Figure 6 (a) shows the experimental device using terahertz wave to detect coffee beans, but the display of each part of the experimental device in the picture is not clear. We hope to add a schematic diagram of the device to fully demonstrate its working principle.
- The artificial intelligence model you proposed in this paper can be applied not only to the selection of coffee beans, but also to other practical sorting fields. Please ask, I hope to give some detailed examples.
Some examples of grammatical errors, spelling errors, and unclear sentence structures, can be modified for better clarity.
Comments on the Quality of English Languagen
Author Response
- We thank the reviewer for their comment. The lack of crystallization enables many more relaxation channels than for a single crystalline material, resulting in a shorter lifetime: a common effect in solid-state physics.
- As mentioned in the Methods section, quaker beans are split before roasting. A 5 mm bean, when split in two, results in two pieces slightly larger than 2 mm each.
- We have proposed training AI models on the specific dispersion features of coffee beans that we have identified. However, our work does not focus on developing or testing these models.
- We appreciate the suggestion of the reviewer and have added to Fig. 6 a schematic of the optical head (now panel b), adapted the caption text as well as the main text to both include this new panel.
- As mentioned in the previous comment, this work focuses on the potential of THz technology for coffee bean sorting, not on the use of AI models. Therefore, discussions on modern AI models for THz applications will be addressed in our upcoming manuscripts.
Round 2
Reviewer 1 Report
Comments and Suggestions for Authors
Most of the issues has been addressed, although clearly with as little effort as possible. That being said, the manuscript is now publishable in it present form.
Author Response
We thank the reviewer, and would like to point out to the editor the statement of the reviewer "the manuscript is now publishable in it present form"
Reviewer 3 Report
Comments and Suggestions for Authors
There are some issues with the reference 5.
- We recorded zero transmission in the range 400 − 8000 cm−1 for a green coffee bean slab polished to 0.1 mm using a Vertex 70 FTIR spectrometer. Calibration measurements indicate that T < 10−3 , leading to an estimated penetration depth of less than 40 µm.
Author Response
This is a new comment, not included in the first round of review. I am not sure this agrees with the journal policies.
Furthermore, the stated reference [5] is fully correct: zero transmission was recorded: calibration measurements indicate this corresponds to a transmission level of < 1E-3.